# Internalizing and externalizing mental health problems affect in-school adolescent's health-related quality of life in eastern Ethiopia: A cross-sectional study

**Gari Hunduma**[1]*, **Yadeta Dessie**[2], **Biftu Geda**[3], **Tesfaye Assebe Yadeta**[1], **Negussie Deyessa**[4]

**1** School of Nursing and Midwifery, College of Health and Medical Sciences, Haramaya University, Harar, Ethiopia, **2** School of Public Health, College of Health and Medical Sciences, Haramaya University, Harar, Ethiopia, **3** Department of Nursing, School of Health Sciences, Madda Walabu University, Robe, Ethiopia, **4** Department of Preventive Medicine, School of Public Health, College of Health Sciences, Addis Ababa University, Addis Ababa, Ethiopia

* garihunduma@gmail.com

**Data Availability Statement:** All relevant data are within the manuscript and its Supporting Information files.

## Abstract

### Aims

This study aimed to examine the association between mental health problems and health-related quality of life (HrQoL) among in-school adolescents 13–19 years in the Harari region, eastern Ethiopia.

### Materials and methods

A cross-sectional study was conducted on 3227 in-school adolescents aged 13 to 19 using multistage sampling. The KIDSCREEN-10 questionnaire assessed health-related quality of life (HrQoL), while a self-administered version of the strength and difficulty questionnaire (SDQ) examined mental health issues. Data were double entered, validated, and cleaned using EpiData version 3.1 and analyzed using STATA 14.1. An ordinal logistic regression model investigated the link between the outcome variable and the predictors. The results were reported using an odds ratio with a 95% confidence interval (CI), and a p-value of less than 0.05 was considered statistically significant.

### Results

A quarter of the adolescents (23%) reported poor health-related quality of life; adolescents with internalizing and externalizing mental health problems had the lowest health-related quality of life. After controlling for potential confounders, adolescents with abnormal (AOR = 0.48, 95% CI: 0.39, 0.59) and borderline (AOR = 0.59, 95% CI: 0.45, 0.78) levels of internalizing problems had a 52% and 41% lower probability of having high HrQoL than those with normal levels. Furthermore, individuals with abnormal (AOR = 0.59, 95% CI: 0.45, 0.77) and borderline (AOR = 0.64, 95% CI: 0.45, 0.92) levels of externalizing difficulties had a 41% and 36% lower chance of having a high health-related quality of life.

**Funding:** After successfully defending the study proposal, this work was sponsored by the University of Haramaya Scientific Research Grant number (HURG-2020-02-01-92). The funder had no involvement in the study's design, data collection, analysis, interpretation, or the manuscript's writing.

**Competing interests:** The authors have declared that no competing interests exist.

## Conclusions

Nearly a quarter of in-school adolescents had poor health-related quality of life. High scores for internalizing and externalizing mental health problems significantly impacted the adolescents' health-related quality of life. This emphasizes the need to address mental health issues in the school setting to improve adolescents' overall quality of life.

## Introduction

Adolescents' mental health issues are becoming a significant global public health concern, significantly burdening the existing healthcare system [1,2]. Mental illness negatively affects adolescents, their parents, and surrounding families, especially regarding health-related quality of life (HrQoL) [3,4]. Mental health disorders encompass a broad spectrum of behaviors, including internalizing (inability to control negative feelings) and externalizing (issues with regulating unwanted conduct) difficulties marked by a mix of dysfunctional emotions, behavior, and interpersonal connections [5,6]. Half of all adult mental health disorders begin during adolescence [7], a period marked by rapid physical, emotional, and cognitive changes [8]. People's everyday functioning, social interactions, and HrQoL are all affected by mental health concerns [9]. Poor adolescent mental health is associated with lower HrQoL [10], lower educational achievement [11], and a higher likelihood of engaging in risky behaviors, such as substance misuse, self-harm, suicide attempts, and suicide [12].

HrQoL is a multifaceted concept incorporating subjective factors of physical and psychological (emotional and cognitive) functioning, independence, personal beliefs, social functioning, and well-being [13,14]. People with a low HrQoL are less likely to develop and mature into healthy adults [15]. Therefore, HrQoL in adolescents can be used to predict healthcare costs and illness [16]. Although HrQoL is commonly used to influence health and welfare policy, it is understudied in community settings, especially among adolescents experiencing mental health issues at an increasing rate [17,18]. Therefore, a better understanding of the link between mental health issues and health-related quality of life in adolescents is a top priority, with significant implications for improving HrQoL in the face of mental health issues [2].

However, in low-income countries like Ethiopia, understanding the relationship between mental health problems and HrQoL in the teenage population is still lacking. In most countries around the world, ensuring a good life for all members of the people, especially adolescents, is a top priority. However, studies on adolescents' perception of their HrQoL and the link to mental health problems are scarce, and information is entirely based on data from higher-income countries [19]. Most studies have evaluated the HrQoL of individuals with physical condition rather than mental health conditions [20]. Additionally, more HrQoL studies have targeted adults than adolescents [21]. In the studies involving adults, those with mental disorders consistently reported lower HrQoL than healthy controls [22,23]. For example, a community-based cross-sectional study conducted in Kenya showed that the quality of life among adolescents with clinical mental health problems was lower than that of adolescents with normal mental health problems [24].

In Ethiopia, adolescents with mental health problems have been neglected in HrQoL studies. In addition, a few clinical-based studies have analyzed the relationship between HrQoL and chronic physical diseases, such as Type 1 diabetes[25], epilepsy [26], overweight [27], and other chronic illnesses [28]. However, no studies have evaluated the relationship between HrQoL and adolescent mental health problems.

Therefore, it is essential to examine the direction and strength of the association of HrQoL with mental health problems among in-school adolescents. Moreover, community-based studies among adolescents are needed to better understand the relationships between mental health and HrQoL with multiple health behaviors and socioeconomic confounders. To the best of the authors' knowledge, no study has investigated the association of mental health problems with the quality of life of in-school adolescents in Ethiopia. Thus, this study aimed to examine the associations between mental health problems and HrQoL among in-school adolescents 13–19 years in the Harari region, eastern Ethiopia.

## Materials and methods

### Study period, settings and design

From November 24 to December 31, 2020, a school-based cross-sectional survey was undertaken in the Harari region of eastern Ethiopia, 511 kilometers from the capital Addis Ababa. Harar city, the region's capital, is home to the bulk of the region's inhabitants (54.2%) [29]. Psychoactive substances, such as *Khat* (Catha edulis), tobacco, and coffee play a significant role in the region's commercial operations [29]. Primary (7th - 8th grades) and secondary (9th - 12th grades) schools are included in the study.

### Population and sampling

The study population was adolescents from randomly selected schools; the source population was all in-school adolescents in the region. Eligible schools and adolescents were chosen using a multistage stratified sampling method. Twenty-three public and private schools from urban and rural locations were randomly selected using a lottery technique. Finally, from each school, sections were chosen randomly from each grade level using a lottery approach based on the number of sections and students included.

The sample size was calculated using OpenEpi software using sample size assumptions for comparing two means and a formula for the difference in means. According to a Kenyan study, the mean and standard deviation of HrQoL among persons with mental health disorders was 71.2 (±15.8) [24]. The computed minimum sample size for this study was 1003 based on the assumptions of a mean difference of 3, 95% confidence level, power of 80%, allocation ratio of 1:1, a design effect of 2, and a 15% non-response rate. However, data were collected from 3227 adolescent students participating in a large school-based mental health study.

### Data collection

A structured and standardized guided self-administered questionnaire was used for collecting the data. The questionnaire included items on sociodemographic variables, psychosocial factors, behavioral factors, adolescent mental health problems, and HrQoL. The questionnaires were initially developed in English and translated into Amharic and Afan Oromo, the region's two most widely spoken languages. The backward and forward translation technique was utilized to maintain uniformity throughout translations. The final version was reviewed by mental health professionals and English, Amharic, and Afan Oromo language experts. The questionnaire was improved and conceptualized based on the feedback of these experts. Before the primary data collection, pretests were undertaken, the Cronbach alpha for reliability and validity was checked, and the report was deemed acceptable. The data was collected at the adolescent students' schools. For those students whose rooms were unsuitable for completing the questionnaire, an appropriate and fit setting (rooms) was arranged. The maximum classroom size was reduced to 25 students due to the COVID-19 pandemic, thus providing a favorable

location for students to discreetly complete the questionnaire. To ensure data quality, data collectors provided the students an orientation on the study's goals and how to complete the questionnaire. Two data collectors were assigned per session to assist and guide the students.

## Variables and measurements

The quantity and type of properties owned by the respective households were used to calculate a wealth index reported by the adolescents, then evaluated using the principal component analysis [30]. Substance use (whether they used alcohol, smoked cigarettes, or chewed *Khat* in the previous month) was measured using nine questions from prior research. Additionally, adolescents were asked about the presence of parental mental illness.

Mental health problems (internalizing/emotional and externalizing/behavioral problems) were assessed using the strength and difficulty questionnaire (SDQ-25) [31]. The SDQ has 25 items categorized into five sub-scales, with five items each measuring conduct, hyperactivity, emotional issues, peer relationship problems, and prosocial behaviours. Each item was answered on a three-point Likert scale ranging from 'not true' (rated 0), 'somewhat true' (rated 1), to 'certainly true' (rated 2). 'Somewhat true' was always noted as 1, but 'not true' and 'certainly true' varied depending on the scale elements [32]. The SDQ was scored using the predictive algorithm converted into Stata syntax available on the SDQ information website [33,34]. Higher scores on the SDQ scale are associated with a greater risk of mental health problems. By applying the method of score banding reported by Goodman, the SDQ total difficulties scores were categorized into 'normal' (0–15), 'borderline' (16–19), and 'abnormal' (20–40) scores [32].

We summed the first four categories (excluding the prosocial behavior items), generating a total difficulty score ranging from 0 to 40. The sum of the conduct and hyperactivity scales was used to create externalizing scores, ranging from 0 to 20; the sum of the emotional and peer problem scales generated the internalizing scores, ranging from 0 to 20. The internalizing problem subscale category was normal (0–7), borderline (8), and abnormal (9–20), while the externalizing subscale for normal, borderline, and abnormal was (0–8), (9), (10–20), respectively [35,36]. This study considered a borderline category score the cut-off point for each difficulty sub-score, indicating mental health problems. The Cronbach's α for SDQ total was 0.764, while it was 0.55 and 0.65 for the internalizing and externalizing scores in the current sample, respectively.

Health-related quality of life (HrQoL) was assessed using the self-reported versions of the KIDSCREEN-10 Index [37], which is used for children and adolescents aged 8–18 years. The index included ten items, a short version of the KIDSCREEN-27 [38,39] questionnaire for health surveys and epidemiological studies. The index consists of questions on physical well-being, psychological well-being, autonomy and relationships with parents, peers and social support, and school environment. The KIDSCREEN-10 item statements were: (1) Have you felt fit and well? (2) Have you felt full of energy? (3) Have you felt sad? (4) Have you felt lonely? (5) Have you had enough time for yourself? (6) Have you been able to do the things that you want to do in your free time? (7) Have your parent(s) treated you fairly? (8) Have you had fun with your friends? (9) Have you got on well at school? (10) Have you been able to pay attention? The possible responses for statements one and nine were not at all, slightly, moderately, or very-extremely. For the remaining items, the response categories included: never, seldom, quite often, very often, or always.

The KIDSCREEN-10 index scores were calculated by adding the scores from each question and then assigning Rasch person parameters (PP) to each potential sum score. We used the multinational European sample to calculate mean values and standard deviations. T-values

with a mean of 50 and a standard deviation (SD) of 10 were created from the index score. Higher scores imply a higher level of HrQoL. Based on the T-scores from the KIDSCREEN-10 index, the adolescents were divided into three groups: low HrQoL, defined as individuals scoring below the 1st percentile (scores below 38.75) on the KIDSCREEN-10 index, medium HrQoL was defined as individuals scoring between the 2nd and 3rd percentile (scores 38.75 to 57.30), and high HrQoL defined as individuals scoring above the 3rd percentile (scores above 57.30). The Cronbach's α of the KIDSCREEN-10 index for the overall sample was 0.7414 in the current study.

## Statistical analysis

The data were double entered, validated, and cleaned using EpiData 3.1 and analyzed using STATA 14.1. Descriptive statistics of the categorical data were summarized using the Chi-square test. Those variables with a p-value < 0.20 in the bivariate analyses were considered in the multivariable analysis. The association between mental health and health-related quality of life was investigated using stepwise multiple ordinal logistic regression models (OLRMs). Model 1 adjusted individual-related confounding variables to show the link between internalizing and externalizing difficulties and health-related quality of life, and model 2 adjusted for parental-related confounding variables to show the connection between internalizing and externalizing difficulties and health-related quality of life. Model 3 adjusted environmental-related confounding variables to demonstrate the link between internalizing and externalizing difficulties and health-related quality of life. Model 4, the final model, adjusted all confounding variables to illustrate the association between internalizing and externalizing difficulties and health-related quality of life. The OLR model was supported by the parallel lines assumption and the brant test in each case. The outcome was presented as an odds ratio with a 95% confidence interval. Finally, a statistically significant association was set at a p < 0.05.

## Ethical approval

The Institutional Health Research Ethics Review Committee (IHRERC) of Haramaya University's College of Health and Medical Sciences granted ethical permission with ref. No IHRERC/149/ 2019. Respondents, parents/guardians, and school administrators were given complete and accurate information about the study, including the purpose, procedures, risks, and benefits. In addition, the adolescents have been informed their participation in the study was entirely voluntary, that not participating would have no negative consequences for their families or the adolescents, and that they could stop at any point or skip questions they did not want to answer.

For participants aged 13 to 17, we obtained a written voluntary assent from the adolescents and a written informed and signed voluntary consent from one of their parents or guardians. Participants aged 18 and up signed a consent form. The written questions not included personal identifiers to protect the participants' privacy. All of the information gathered was anonymized and stored on a password-protected computer. Everything was kept private and anonymous. The participants and their parents were informed that the acquired information would only be released to assist in knowledge generation. The study followed the Declaration of Helsinki's ethical principles for medical research involving human subjects.

## Results

### Sociodemographic characteristics

A total of 3227 adolescent students were included in the study. The mean age of the respondents was 15.69 (SD±1.79) years, ranging from 13 to 19 years. The majority (83.85%) of the

respondents were from urban and half (51.75%) were girls, 51.84% learn in primary schools and 54.2% were Muslims. Seven hundred forty (22.93%) had one or more mental health problems. The proportion of mental health problems by subscale was 24.17% for internalizing and 11.93% for externalizing problems (**Table 1**).

## Association between HrQoL and mental health problems among adolescents

The overall mean score of health-related quality of life (HrQoL) among adolescent students was 48.2 (95% CI: 47.71, 48.7). The mean was 50 (95% CI: 49.5, 50.6) among adolescents with low total SDQ scores and 42 (95% CI: 41, 42.9) among adolescents with high total SDQ scores. The difference was statistically significant (t = 14.3973, p < 0.001). Furthermore, the quality of life of adolescents was categorized into three levels (low, medium, and high). Accordingly, of

**Table 1. Socio-demographic characteristics and mental health status of the respondents in Harari regional state, Eastern Ethiopia, 2020, (N = 3227).**

| Variables | Categories | Frequency | Percentage |
|---|---|---|---|
| **Age** | 13–15 | 1,526 | 47.29 |
| | 16–19 | 1,701 | 52.71 |
| **Sex** | Male | 1557 | 48.30 |
| | Female | 1670 | 51.70 |
| **Residence** | Urban | 2,706 | 83.85 |
| | Rural | 521 | 16.15 |
| **School type** | Public | 2,162 | 67.00 |
| | Private | 1,065 | 33.00 |
| **Parental marital status** | Living together | 2,453 | 76.01 |
| | Living separately | 240 | 7.44 |
| | Divorced/Separated/widowed | 534 | 16.55 |
| **Family size** | $\leq 3$ | 418 | 12.95 |
| | 4–7 | 2,156 | 66.81 |
| | $\geq 8$ | 653 | 20.24 |
| **Wealth index** | Lowest | 1,291 | 40.01 |
| | Middle | 1,295 | 40.13 |
| | Highest | 641 | 19.86 |
| **Alcohol use** | Never use | 2,864 | 88.75 |
| | Ever use | 363 | 11.25 |
| **Tobacco use** | Never use | 3,042 | 94.27 |
| | Ever use | 185 | 5.73 |
| *Khat use* | Never use | 2,712 | 84.04 |
| | Ever use | 515 | 15.96 |
| **History of parental mental illness** | No | 2,814 | 87.20 |
| | Yes | 413 | 12.80 |
| **Any chronic medical conditions** | No | 2,549 | 78.99 |
| | Yes | 678 | 21.01 |
| **Internalizing problems** | Normal | 2,447 | 75.83 |
| | Borderline | 221 | 6.85 |
| | Abnormal | 559 | 17.32 |
| **Externalizing problems** | Normal | 2,842 | 88.07 |
| | Borderline | 121 | 3.75 |
| | Abnormal | 264 | 8.18 |

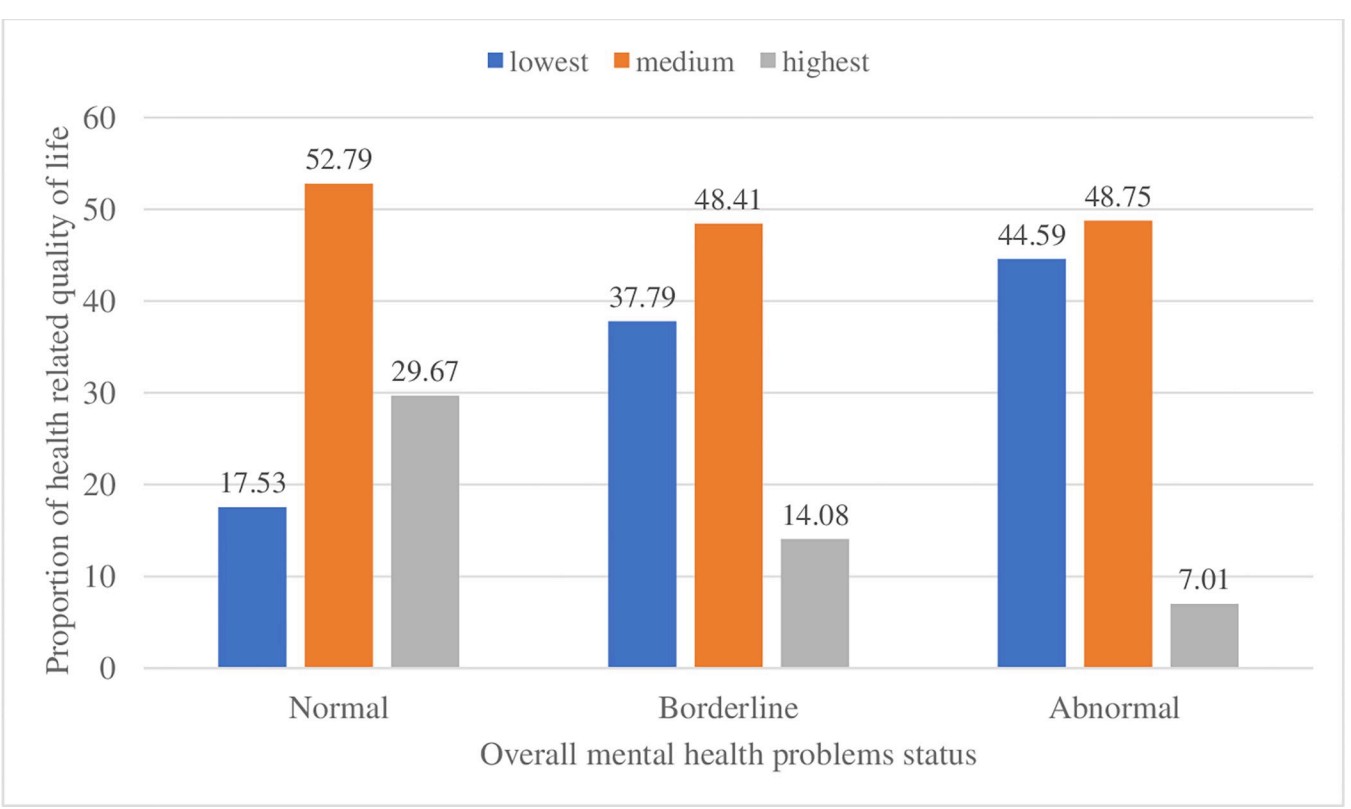

**Fig 1. Levels of HrQoL with adolescents' mental health problem status in Harari region, eastern Ethiopia 2021.**

the 3227 participants, 737 (23%), 1670 (52%), and 820 (25%) of adolescents were at low, medium, and high HrQoL, respectively. Data analysis revealed that the quality of life was lower among adolescents with high mental health problem scores across all dimensions (Fig 1).

The multivariable ordinal logistic regression (Table 2) showed that age, residence, parental marital status, family size, household wealth index, alcohol use, history of parental mental illness, and chronic medical conditions were significantly associated with health-related quality of life among in-school adolescents.

For adolescents with abnormal (AOR = 0.48, 95% CI: 0.39, 0.59) and borderline (AOR = 0.59, 95% CI: 0.45, 0.78) levels of internalizing problems, the odds of having a high HrQoL decreased by 52% and 41%, respectively compared to those with normal levels. Furthermore, adolescents who had abnormal (AOR = 0.58, 95% CI: 0.44, 0.75) and borderline (AOR = 0.63, 95% CI: 0.43, 0.91) levels of externalizing problems, the odds of a high HrQoL decreased by 40% in both groups compared to those with normal levels, holding all other variables constant.

From those controlled variables, the estimated adjusted odds ratio indicated that those 16–19 years old (AOR = 0.70, 95% CI: 0.60, 0.80) were 0.70 times less likely to have high levels of HrQoL compared to the age category 13–15. Likewise, compared to lowest, the highest HrQoL was less likely among rural residence (AOR = 0.50, 95% CI: 0.40, 0.80), separated and divorced parents (AOR = 0.60, 95% CI: (0.50, 0.80) and (AOR = 0.80, 95% CI: (0.60, 0.90), large family size (≥8) (AOR = 0.70, 95% CI: 0.50 to 0.90), alcohol consumption (AOR = 0.80, 95% CI: (0.60, 0.90), having history of parental mental illness (AOR = 0.60, 95% CI: 0.50, 0.80), and

**Table 2. Ordinal logistic regression analyses showing the association between variables or mental health problems and HrQoL among In-School Adolescents in Harari Region, Eastern Ethiopia, 2020, (n = 3227).**

| Variables | HrQoL: OLR (95% CI) | | | |
|---|---|---|---|---|
| | Model 1 [a] | Model 2 [b] | Model 3 [c] | Model 4 [d] |
| **Internalizing problems** (Ref Normal) | Ref | Ref | Ref | Ref |
| Borderline | 0.50 (0.40, 0.70) * | 0.60 (0.40, 0.70) * | 0.50(.40, 0.70) * | 0.60 (0.50, 0.80) * |
| Abnormal | 0.40 (0.30, 0.50) * | 0.50 (0.40, 0.60) * | 0.40 (0.30, 0.50) * | 0.50 (0.40, 0.60) * |
| **Externalizing problems** (Ref Normal) | Ref | Ref | Ref | Ref |
| Borderline | 0.60 (0.40, .80) * | 0.60 (0.40, 0.90) * | 0.60 (0.40, 0.80) * | 0.60 (0.50, 0 .90) * |
| Abnormal | 0.50 (0.40, 0.70) * | 0.50 (0.40, 0.70) * | 0.50 (0.40, 0.70) * | 0.60 (0.45, 0.80) * |

[a] The model included individual-related covariates that were significant at p < 0.05 (age, sex, alcohol use, cigarette smoking, khat chewing and any chronic physical illness).

[b] The model included family- related covariates that were significant at p < 0.05 were included I the model (parental marital status, family size and history of mental illness in the family).

[c] The model included environmental-related covariates that were significant at p < 0.05 (residence, school type and house hold wealth index).

[d] All covariates significant at p < 0.05 were included in the model (age, sex, cigarette smoking, any chronic physical illness, parental marital status, family size and history of mental illness in the family, residence, school type and house hold wealth index) (S1–S4 Tables).

OLR: ordinal logistic regression, HrQoL: health-related quality of life, ref: reference group, *values indicate statistical significance (p < 0.05).

having any chronic medical illness (AOR = 0.80, 95% CI: 0.70, 0.90). Whereas the highest HrQoL was more likely among adolescents of families in the middle (AOR = 1.40, 95% CI: 1.20, 1.60) and high (AOR = 1.40, 95% CI: 1.20, 1.70) wealth index compared to the lowest.

## Discussion

This study was conducted to assess the link between mental health problems and HrQoL among adolescents attending school in the Harari region of eastern Ethiopia. The results indicate that approximately one in four adolescents (23%) had a low HrQoL. The multivariate ordinal logistic regression analyses showed that elevated scores of self-reported internalizing and externalizing mental health problems are independently and significantly associated with the level of HrQoL. This study also explored the role of other covariates linked to adolescence and family or parents that attenuate the relationship between the two. We found that poor HrQoL was associated with age, experiencing one or more chronic physical illnesses, residing in rural areas, and alcohol consumption. The parenteral characteristics associated with decreased HrQoL in adolescents included divorced or living separately, large family size ($\geq 8$), lower household wealth index, and family history of mental illness.

The lower HrQoL was strongly associated with the presence of borderline to an abnormally wide range of internalizing and externalizing mental health problems. The HrQoL was reduced by half among those with borderline and abnormal internalizing problems and a 40% reduction among those externalizing problems. This finding is supported by previous evidence in Nigeria [40], Kenya [41], Rotterdam [42], Australia [43], and in the United States [44], which revealed that the HrQoL of adolescents with mental health problems is strongly affected and functionally inferior even in comparison with adolescents with other health problems. One possible explanation for the strong association of internalization and externalization of mental health problems on HrQoL is that the internalization and externalization of mental health problems affect the functioning of adolescents in multiple ways.

Adolescents with internalized and externalized mental health problems may have an exaggerated psychological, emotional [41] and physical disability and maladjustment [45]. For instance, adolescents with mental health problems are more likely to be absent from school,

affecting their academic performance, and be rejected by their peers, which affects their social functioning. Therefore, adolescents with borderline and abnormal SDQ scores may feel more dissatisfaction, rejection, apathy, and disappointed in their everyday lives than other adolescents.

This finding highlights the importance of considering mental health issues as significant predictors of HrQoL in Ethiopian adolescents. In addition, local mental health practitioners may use this finding to obtain information on the potential burden of mental health problems throughout the spectrum of problems affecting adolescents referred to their practice. This is important to identify the groups of adolescents who need the most support.

The results suggest that HrQoL is primarily affected in adolescents with problems of internalizing rather than externalizing. However, past research has also shown that internalizing problems are more related to the quality of life than externalizing problems among adolescents [41,46]. The reason may be that issues of externalization have a greater impact on family members than on adolescents themselves, compared with issues of internalization [47,48]. In addition, adolescents with externalizing behavioral issues may not experience their symptoms as problematic, which may well explain our findings.

Current results revealed that HrQoL tends to decrease with age in all adolescents in the sample. Still, the results show that this effect is less pronounced in adolescents with externalizing problems. A previous study also reported similar results regarding the effect of age moderation on the relationship between externalizing issues and HrQoL in adolescents [46,49]. The reason could be the impact of prominent externalizing problems among younger adolescents than older ones, so the quality of life of younger adolescents is more affected. On the other hand, older adolescents are more aware of their challenges than younger adolescents and more likely to perceive that they are different from their peers and applying for help can have less impact on their quality of life. This might influence their report of HrQoL. In this study, gender does not affect the relationship between mental health problems and decreased quality of life among adolescents. These findings are inconsistent with previous studies that have shown that the impact of psychopathology on quality of life is greater for girls than for boys [50]. Studies have identified an association between the seriousness of mental health issues and gender, revealing the impact of mental health issues on quality of life are more significant in girls than boys. These studies argued that boys had far more externalizing behavioral problems than girls [51] and adolescents with externalizing behavioral problems may not experience their symptoms as problematic.

The results showed that all chronic physical conditions were associated with poor HrQoL in adolescents. Previous studies have also reported that any chronic physical disorder affects HrQoL in adolescents [25,43]. The potential explanation for this association could be chronic physical disease can have a broad impact, including HrQoL, which measures the impact of a young person's health on their everyday emotional, social and physical functioning [43]. The impact can also affect their academic performance, limit social activities with peers, and reduce activities at home and with the family [52].

A history of parental mental illness reported by adolescents is associated with a consistent reduction in HrQoL in adolescents with all other variables. Previous research has also revealed maternal psychopathology is associated with poorer HrQoL in children and adolescents [53]. Contrary to this, other studies have reported that parental mental health problems may not be associated with HrQoL deficits in adolescent boys [54]. However, parents' mental health issues can be significant and potentially modifiable, so they should be targeted for intervention aimed at improving the quality of life of adolescents [53] further investigation is warranted.

### Strength and limitations of the study

The survey includes urban and rural students from public and private schools, which may minimize the major socioeconomic confounding factors. The study also employed standardized tools that were guided and self-administered. Yet there are certain limitations to this study. The study used a cross-sectional design that may not preclude the possible inverse causality between mental health and HrQoL and vice versa.

Consequently, future studies should examine the same associations as this study on the same sample, with a prospective study design, to answer the question of the inverse causation between the variables. Another limitation of this study is it analyzed HrQoL in terms of the level of perception of students, which could be affected by several characteristics of students. Furthermore, the sample was taken only from schooled adolescents and did not represent the out-of-school population. Therefore, epidemiologically, it is necessary to study a representative sample of the community that best represents the population as a whole.

## Conclusion

Overall, about one-fourth of in-school adolescents have a poor health-related quality of life. A high rating of internalizing and externalizing mental health problems has a significant role in the adolescent quality of life, which is principal in overall adolescent development. This association highlights the need for screening, early detection, and treatment of internalizing and externalizing problems among school adolescents to improve health-related quality of life.

## Supporting information

**S1 Table. Ordinal logistic regression analyses showing the association between individual related variables, mental health problems and HrQoL among in-school adolescents in Harari Region, Eastern Ethiopia, 2020, (n = 3227).**
(DOCX)

**S2 Table. Ordinal logistic regression analyses showing the association between environmental related variables, mental health problems and HrQoL among in-school adolescents in Harari Region, Eastern Ethiopia, 2020, (n = 3227).**
(DOCX)

**S3 Table. Ordinal logistic regression analyses showing the association between family related variables, mental health problems and HrQoL among in-school adolescents in Harari Region, Eastern Ethiopia, 2020, (n = 3227).**
(DOCX)

**S4 Table. Analyses based on ordinal logistic regression family-related characteristics, mental health concerns, and HrQoL among in-school adolescents in Harari Region, Eastern Ethiopia, 2020 (n = 3227).**
(DOCX)

**S1 Dataset.**
(DTA)

## Acknowledgments

We want to acknowledge Haramaya University for giving us the approval to conduct this research. We also like to express our gratitude to the Harari Region Education Office, which brought together school directors, instructors, and participants. Our sincere appreciation also

goes to data collectors for painstakingly arranging and carrying out the work. Finally, we also sincerely thank Tara Wilfong for her excellent assistance in helping to revise the manuscript's language and grammar.

## Author Contributions

**Conceptualization:** Gari Hunduma, Yadeta Dessie, Tesfaye Assebe Yadeta, Negussie Deyessa.

**Data curation:** Gari Hunduma, Yadeta Dessie, Biftu Geda, Tesfaye Assebe Yadeta, Negussie Deyessa.

**Formal analysis:** Gari Hunduma, Negussie Deyessa.

**Investigation:** Gari Hunduma, Negussie Deyessa.

**Methodology:** Gari Hunduma, Yadeta Dessie, Biftu Geda, Tesfaye Assebe Yadeta, Negussie Deyessa.

**Project administration:** Gari Hunduma.

**Software:** Gari Hunduma, Biftu Geda, Negussie Deyessa.

**Supervision:** Gari Hunduma, Tesfaye Assebe Yadeta.

**Validation:** Gari Hunduma, Yadeta Dessie, Biftu Geda, Negussie Deyessa.

**Writing – original draft:** Gari Hunduma.

**Writing – review & editing:** Yadeta Dessie, Biftu Geda, Tesfaye Assebe Yadeta, Negussie Deyessa.

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
