## [Decision Letter · Decision Letter 0]

17 Jun 2022

PONE-D-22-10505High scores of both internalizing and externalizing mental health problems were the major factors affecting in-school adolescent’s health-related quality of life in eastern Ethiopia: A cross-sectional study

PLOS ONE

Dear Dr. Hunduma,

Thank you for submitting your manuscript to PLOS ONE. After careful consideration, we feel that it has merit but does not fully meet PLOS ONE’s publication criteria as it currently stands. Therefore, we invite you to submit a revised version of the manuscript that addresses the points raised during the review process.

A rebuttal letter that responds to each point raised by the academic editor and reviewer(s). You should upload this letter as a separate file labeled 'Response to Reviewers'.A marked-up copy of your manuscript that highlights changes made to the original version. You should upload this as a separate file labeled 'Revised Manuscript with Track Changes'.An unmarked version of your revised paper without tracked changes. You should upload this as a separate file labeled 'Manuscript'

We look forward to receiving your revised manuscript.

Kind regards,

Sebsibe Tadesse

Academic Editor

PLOS ONE

"Haramaya University funded this work, which the authors gratefully acknowledge. We also like to express our gratitude to the Harari Region Education Office, which brought together school directors, instructors, and participants. Finally, we'd like to thank the data collectors for painstakingly arranging and carrying out the work"

 "After successfully defending the study proposal, this work was sponsored by the University of Haramaya Scientific Research Grant number (HURG-2020-02-01-92). The funder had no involvement in the study's design, data collection, analysis, interpretation, or the manuscript's writing."

Reviewers' comments:

Reviewer's Responses to Questions

**Comments to the Author**

1. Is the manuscript technically sound, and do the data support the conclusions?

Reviewer #1: Yes

Reviewer #2: Yes

2. Has the statistical analysis been performed appropriately and rigorously? 

Reviewer #1: Yes

Reviewer #2: Yes

3. Have the authors made all data underlying the findings in their manuscript fully available?

Reviewer #1: Yes

Reviewer #2: No

4. Is the manuscript presented in an intelligible fashion and written in standard English?

Reviewer #1: Yes

Reviewer #2: No

5. Review Comments to the Author

Reviewer #1: The authors describe a cross-sectional study with the objective to examine the associations between mental health problems and health-related quality of life among school adolescents aged 13–19 years, in the Harari region, eastern Ethiopia.

Topic of interest to readers. The manuscript it is easy to follow, is well written and provides an important contribution to the field and the conclusions agree with the data presented.

Discussion is clear and nicely places study findings in context with other studies. The findings are certainly of interest to the readers of Plos One.

Nevertheless, I have some minor issues below for authors to consider.

When an abbreviation is used for the first time, you should use it throughout all the text. This happens with health-related quality of life (HrQoL).

Abstract - Please add the sample size.

Methods - Please provide the Cronbach's alpha results for reliability and validity.

Line 330 – a reference is need to support the sentence.

Reviewer #2: Thank you for the opportunity to review this article. It includes a very significant number of adolescents from Ethiopia using comprehensive questionnaires and dealing with a significant public health problem - mental health.

Here are some suggestions that would require authors' attention.

1) Abstract seem to exceed the word limit 300 words. Methods section could be easily shortened. Also title is very long and it would benefit from cutting down words.

2) In the abstract and throughout the manuscript you use words abnormal and normal to describe mental health (MH) status. I strongly suggest that you change the reference to normal/abnormal since public health and medical literature has largely stop using them since abnormal reference stigmatizes people with MH problems.

3) Although the writing of the paper is in general in good English there are several sentences and occasions where there is need for improvement. I suggest that you use some professional to proofread the next version.

4) Methods; you had a section from khat problem in Ethiopia. However, you didn't discuss it in your discussion. Unless you think it is very important issue, you may delete most of that text.

5) I didn't see anything related to translation or cultural adaptation of the instruments in Ethiopia. This is a very important issue, since the validity and reliability of the instrument is related to translation process. If those processes have been done already just give reference. Also in the discussion you should discuss the potential cultural issues related the KIDSCEEN-10. It is possible that not all items are not that good indicators for all kids, like working at home or outside home may be more common in your culture and it may impact on several items not being comparable to European data that you use as a reference.

6) History of MH in family variable; I was not sure is it relevant or how to interpret it. You are already measuring MH in your study so you don't need a proxy for MH problems through family. If you intend to measure hardship of the family (mental, social and economic struggles) then you can discuss from that point of view. In addition in discussion (lines 343-349) you discuss parental MH while your question is wider family MH.

7) In results, conclusion and discussion you refer to 23% of having low HRQoL. It is trivial since you divide you sample to quartiles so by definition you will have about 1/4th low level kids. Related to that, Figure 1 is not needed.

8) Fig 2; hard to understand your groupings. I would rather like to see how HRQoL levels are within MH categories. Improve the graphics quality.

9) I found 62 references a high number, you could delete some less important references.

10) In the results you report regression results twice in the same sentence. Just simplify so that you don't also duplicate all the results from tables. Use the same level of decimals in the text and tables.

11) As I mentioned above, you don't have any discussion about the KIDSCREEN-10 instrument performance and interpretation in Ethiopia. Has other instruments been used in your country, and are the results comparable? Was there much missing values?

6. PLOS authors have the option to publish the peer review history of their article (what does this mean?). If published, this will include your full peer review and any attached files.

Reviewer #1: No

Reviewer #2: No

---

## [Author Response · Author response to Decision Letter 0]

20 Jul 2022

July 20, 2022

To: PLOS ONE journal Editor-in-chief

Subject: Revision and resubmission of the manuscript (manuscript ID: PONE-D-22-10505) 

Dear Editor and reviewers,

We would like to express our appreciation to the editor and the reviewers for their insightful and constructive comments on our manuscript entitled "High scores of both internalizing and externalizing mental health problems were the major factors affecting in-school adolescent's health-related quality of life in eastern Ethiopia: A cross-sectional study" The comments and suggestions offered by the reviewers have immensely helped us to improve our manuscript.

We have considered every comment and have responded point by point, indicating how we addressed them and tracking the changes we have made. The changes are presented in the revised manuscript. We recommend the Data Availability statement looks like the following: “All relevant data used for this study are within the manuscript and its supporting information file.” Further, the funding statement should be presented in the Funding Statement section of the online submission form as follow: "After successfully defending the study proposal, this work was sponsored by the University of Haramaya Scientific Research Grant number (HURG-2020-02-01-92). The funder had no involvement in the study's design, data collection, analysis, interpretation, or the manuscript's writing." Our responses to the reviewers' comments are provided below.

We hope the revised manuscript is better suited to your esteemed journal and we are happy to consider any further revision as deemed necessary. 

Sincerely,

Gari Hunduma 

I. REPLY TO COMMENTS FROM EDITORS

Comments / suggestions 

Please ensure that your manuscript meets PLOS ONE's style requirements, including those for file naming Throughout the review, we verified the guidelines and adhered to all instructions and criteria.

We note that you have provided funding information that is not currently declared in your Funding Statement. However, funding information should not appear in the Acknowledgments section or other areas of your manuscript.

 We will only publish funding information present in the Funding Statement section of the online submission form. 

 "After successfully defending the study proposal, this work was sponsored by the University of Haramaya Scientific Research Grant number (HURG-2020-02-01-92). The funder had no involvement in the study's design, data collection, analysis, interpretation, or the manuscript's writing."

Thank you for the comments and are well appreciated. We modified the representations based on the comments given, and we welcome the changes you made to the funding statement on our behalf. Our cover letter above contains our modified statements. We accepted your recommendation. "After successfully defending the study proposal, this work was sponsored by the University of Haramaya Scientific Research Grant number (HURG-2020-02-01-92). The funder had no involvement in the study's design, data collection, analysis, interpretation, or the manuscript's writing."

In your Data Availability statement, you have not specified where the minimal data set underlying the results described in your manuscript can be found. PLOS defines a study's minimal data set as the underlying data used to reach the conclusions drawn in the manuscript and any additional data required to replicate the reported study findings in their entirety. All PLOS journals require that the minimal data set be made fully available. "Upon re-submitting your revised manuscript, please upload your study’s minimal underlying data set as either Supporting Information files or to a stable, public repository and include the relevant URLs, DOIs, or accession numbers within your revised cover letter. Any potentially identifying patient information must be fully anonymized. Note that it is not acceptable for the authors to be the sole named individuals responsible for ensuring data access. We will update your Data Availability statement to reflect the information you provide in your cover letter. 

Thank you. The data can be availed; we posted it as supporting information files. The Data Availability statement looks like the following: “All relevant data are within the manuscript and its supporting information file”

Please include your full ethics statement in the ‘M section of your manuscript file. In your statement, please include the full name of the IRB or ethics committee who approved or waived your study, as well as whether or not you obtained informed written or verbal consent. If consent was waived for your study, please include this information in your statement as well. 

Thank you for your valuable concern. We added the entire ethics statement to the methods section of our revised manuscript. We also listed the entire name of the IHRERC that authorized the study protocol.

For participants aged 13 to 17, their parents or legal guardians were given written informed and signed consent; the teenagers were also given written assents. However, those aged 18 and above signed consent forms.

Please include captions for your Supporting Information files at the end of your manuscript, and update any in-text citations to match accordingly. Please see our Supporting Information guidelines for more information 

We appreciate your effort in pointing out what needs to be done better. The amended manuscript now has captions for supporting information files. We made sure that in-text citations correspond to supporting information files. 

Reviewer #1 Evaluation 

The authors describe a cross-sectional study to examine the associations between mental health problems and health-related quality of life among school adolescents aged 13–19 years, in the Harari region, eastern Ethiopia. 

Thank you and accepted as presented.

The topic of interest to readers. The manuscript is easy to follow, is well written, and provides an important contribution to the field and the conclusions agree with the data presented. 

Thank you and accepted as presented.

Discussion is clear and nicely places study findings in context with other studies. The findings are certainly of interest to the readers of Plos One. 

Thank you and accepted as presented.

Nevertheless, I have some minor issues below for the authors to consider. When an abbreviation is used for the first time, you should use it throughout the text. This happens with health-related quality of life (HrQoL). 

Thank you for indicating this area. 

We checked and corrected it throughout the documents. 

Abstract - Please add the sample size. 

We appreciate your point. The abstract makes this clear. kindly refer to line 25.

Methods - Please provide Cronbach's alpha results for reliability and validity. 

Many thanks. The comment is accurate. For reliability and validity, we took into account Cronbach's alpha values. While internalizing and externalizing in the current sample are each at 0.55 and 0.65, respectively. The Cronbach's alpha for the SDQ total is 0.764 (see lines 157). The whole sample's Cronbach's alpha of the KIDSCREEN-10 index for the current study is 0.7414. (See lines 180-181)

Line 330 – a reference is needed to support the sentence. 

Thank you. The comment was accepted, and it was amended. In the updated manuscript, the statement is referenced.

Reviewer #2 Evaluation 

1) Abstract seems to exceed the word limit of 300 words. The methods section could be easily shortened. Also, the title is very long and it would benefit from cutting down words. 

We are grateful, and the comment is acceptable. The updated manuscript abstract is within the allotted word count. Additionally, we revised and deleted a few words from the title.

2) In the abstract and throughout the manuscript you use the words abnormal and normal to describe mental health (MH) status. I strongly suggest that you change the reference to normal/abnormal since public health and medical literature has largely stopped using them since abnormal reference stigmatizes people with MH problems. 

Thank you. The issue brought up is crucial. In the study, stigmatizing terminology should be avoided. However, we used the initial bandings of the SDQ-25 questionnaire cutoff points to classify the adolescent participants' mental health issues as "normal," "borderline," and "abnormal." These bandings were developed using data from a population-based UK survey to select cut-points that would result in an 80 percent "normal," a 10 percent "borderline," and a 10 percent "abnormal" score for children as recommended by Goodman R.(Goodman, 1997). 

The cut-off score was determined by normative banding that categorizes children with the highest 10% score range as abnormal and the second highest 10% as borderline following the original method,

We prefer to maintain the references in their current form because most publications that used the original three banding categories report this way.

 So, to be consistent with the previous publications we prefer this categorization. Some of the publications with a similar report were (Aoki et al., 2021, Addy et al., 2021, Emam et al., 2016, Syed et al., 2007). But we have to consider it in our future studies.

We are happy to consider any further revision as deemed necessary. 

3) Although the writing of the paper is in general in good English there are several sentences and occasions where there is a need for improvement. I suggest that you use some professionals to proofread the next version. 

This comment is acceptable. Using your suggestions as a guide, a language specialist edited the English. Grammar and sentence structure have been fixed in the revised document.

4) Methods; you had a section on the khat problem in Ethiopia. However, you didn't discuss it in your discussion. Unless you think it is a very important issue, you may delete most of that text. 

Most of the text in the method section is deleted because we accept this comment as it is.

5) I didn't see anything related to translation or cultural adaptation of the instruments in Ethiopia. This is a very important issue since the validity and reliability of the instrument is related to translation process. If those processes have been done already just give reference. Also, in the discussion you should discuss the potential cultural issues related the KIDSCEEN-10. It is possible that not all items are not that good indicators for all kids, like working at home or outside home may be more common in your culture and it may impact on several items not being comparable to European data that you use as a reference. 

Thank you for raising this point. The questionnaires were initially developed in English, then translated into Amharic and Afan Oromo, the region's two most widely spoken languages. The backward and forward translation technique was developed to maintain uniformity throughout translations. The final version was reviewed by mental health professionals as well as dual expert for English Amharic and Afan Oromo language experts. The questionnaires were improved and conceptualized based on the feedback of these experts. Before the main data collection, pretests were undertaken, the Cronbach alpha for reliability and validity was checked, and the report was deemed acceptable.

This is described under the data collection section of the method part in the revised manuscript.

6) History of MH in family variable; I was not sure is it relevant or how to interpret it. You are already measuring MH in your study so you don't need a proxy for MH problems through family. If you intend to measure hardship of the family (mental, social and economic struggles) then you can discuss from that point of view. In addition, in discussion (lines 343-349) you discuss parental MH while your question is wider family MH. 

Thank you for bringing up such crucial points. This variable is taken into account as a confounding factor because studies have linked it to children's HrQoL. We questioned teenagers about any mental illnesses that their parents (father, mother, and/or siblings) had (the parental history of mental illness). As a result, the question and discussion are related, and we have fixed it in the updated manuscript.

7) In results, conclusion and discussion you refer to 23% of having low HRQoL. It is trivial since you divide you sample to quartiles so by definition you will have about 1/4th low level kids. Related to that, Figure 1 is not needed.

 Thank you for this point also. It is an acceptable comment. We remove the figure in the revised manuscript. 

8) Fig 2; hard to understand your groupings. I would rather like to see how HRQoL levels are within MH categories. Improve the graphics quality. 

We appreciated your comments. The figure is correctly edited and improved. Since the previous figure 1 is removed now this figure is named figure 1

9) I found 62 references a high number, you could delete some less important references. 

We revised and delete some less important references and the number of references was reduced to 54.

10) In the results you report regression results twice in the same sentence. Just simplify so that you don't also duplicate all the results from tables. Use the same level of decimals in the text and tables. 

We thoroughly revised and corrected this part. The same level of decimals in the text and tables were used in the revised manuscript.

11) As I mentioned above, you don't have any discussion about the KIDSCREEN-10 instrument performance and interpretation in Ethiopia. Has other instruments been used in your country, and are the results comparable? Was there much missing values? 

Thank you. Even though the KIDSCREEN-10 instrument is suitable for children 8 to 18 years, it was not used in Ethiopia, however, as mentioned above, two language experts translated the questionnaire into Amharic and Afan Oromo As per our evaluation during the pre-testing, the translated version was appropriate and conceptually equivalent to the original questionnaire. The report was acceptable after we looked at the items 'Cronbach's α (0.7414). As a result, the target population within the country thought the translations to be culturally appropriate, acceptable, and simple to understand. This might guarantee consistency and comparability between languages, enabling secure worldwide data sharing and country comparisons.

• The fading was also comparable with other instruments used in the country.

• The data has no problem with missing valves

• Additionally, we appreciate the advice to validate the KIDSCREEN instrument's final translation to learn more about the psychometric characteristics of the tool in the target language.

---

## [Editor Report · Decision Letter 1]

25 Jul 2022

Internalizing and externalizing mental health problems affect in-school adolescent’s health-related quality of life in eastern Ethiopia: A cross-sectional study

PONE-D-22-10505R1

Dear Dr. Gari Hunduma,

We’re pleased to inform you that your manuscript has been judged scientifically suitable for publication and will be formally accepted for publication once it meets all outstanding technical requirements.

Kind regards,

Sebsibe Tadesse, PhD

Academic Editor

PLOS ONE

---

## [Editor Report · Acceptance letter]

28 Jul 2022

PONE-D-22-10505R1 

Internalizing and externalizing mental health problems affect in-school adolescent’s health-related quality of life in eastern Ethiopia: A cross-sectional study 

Dear Dr. Hunduma:

I'm pleased to inform you that your manuscript has been deemed suitable for publication in PLOS ONE. Congratulations! Your manuscript is now with our production department. 

Kind regards, 

on behalf of

Dr. Sebsibe Tadesse 

Academic Editor

PLOS ONE